# Preventing Microorganism Contamination in Starting Active Materials for Synthesis from Global Regulatory Agencies: Overview for Public Health Implications

**DOI:** 10.3390/microorganisms13071595

**Published:** 2025-07-06

**Authors:** Francesco Gravante, Francesco Sacchini, Stefano Mancin, Diego Lopane, Mauro Parozzi, Gaetano Ferrara, Marco Sguanci, Sara Morales Palomares, Federico Biondini, Francesca Marfella, Giovanni Cangelosi, Gabriele Caggianelli, Fabio Petrelli

**Affiliations:** 1Local Health Authority of Caserta, San Giuseppe Moscati Hospital, 81031 Aversa, Italy; francesco.gravante@aslcaserta.it; 2Nursing Department Polytechnic, University of Ancona, 60121 Ancona, Italy; francescosacchini@libero.it; 3IRCCS Humanitas Research Hospital, Via Manzoni 56, Rozzano, 20089 Milan, Italy; diego.lopane@hunimed.eu; 4Department of Medicine and Surgery, University of Parma, 43121 Parma, Italy; mauro.parozzi@unipr.it; 5Nephrology and Dialysis Unit, Ramazzini Hospital, 41012 Carpi, Italy; amaranto1984@libero.it; 6A.O. Polyclinic San Martino Hospital, 16132 Genova, Italy; sguancim@gmail.com; 7Department of Pharmacy, Health and Nutritional Sciences (DFSSN), University of Calabria, 87036 Rende, Italy; sara.morales@unical.it; 8Units of Psychiatry, Ast Fermo, 63900 Fermo, Italy; federico.biondini@sanita.marche.it; 9Italian Coordination Volunteer Nurses of Hemergencies Association (Cives), 00184 Rome, Italy; direzioneoperativa@cives-odv.org; 10School of Pharmacy, Experimental Medicine and “Stefani Scuri” Public Health Department, University of Camerino, 62032 Camerino, Italy; fabio.petrelli@unicam.it; 11Azienda Ospedaliera San Giovanni Addolorata, 00184 Rome, Italy; caggianelligabriele@gmail.com

**Keywords:** preventing microorganism contamination, starting active materials for synthesis, global regulatory agencies, public health, overview

## Abstract

Starting Active Materials for Synthesis (SAMS) represents a critical stage in drug manufacturing, directly influencing the microbiological quality and safety of the final product. The introduction of SAMS marks the point where Good Manufacturing Practices (GMP) begin to apply, which are essential for ensuring sterility and preventing microbial contamination during the synthesis process. However, defining the exact point in the process that qualifies as the SAMS is subject to uncertainties, as earlier stages are not always governed by stringent GMP standards. The regulatory differences between various countries further contribute to this issue. This study explores the implications of SAMS selection and use in relation to sterility and infection control, analyzing the guidelines of major Regulatory Authorities and comparing their approaches to GMP. Regulations from several international regulatory agencies were examined, with a particular focus on microbiological control measures and infection protection in the SAMS manufacturing process. The analysis focused on the microbiological control requirements and safety measures applicable to the stages preceding the introduction of SAMS into the production of the final Active Pharmaceutical Ingredients (APIs). Documents published between 2015 and 2025 were included based on predefined criteria regarding relevance, accessibility, and regulatory authority. The analysis revealed significant discrepancies between regulations regarding the definition and management of SAMS. In particular, the regulations in Mexico and India have notable gaps, failing to provide clear guidelines on SAMS sterility and protection against infectious contamination. Conversely, China has introduced risk-based approaches and early-stage microbiological controls, especially for sterile products, aligning with international standards. The European Medicines Agency (EMA), the U.S. Food and Drug Administration (FDA), the Pharmaceutical Inspection Co-operation Scheme (PIC/S), and the World Health Organization (WHO) have well-established systems for microbiological quality control of SAMS, including rigorous measures for the validation of suppliers and risk management to ensure that SAMS does not compromise the microbiological safety of the final product. The regulations in Brazil and Canada introduce additional measures to protect the microbiological quality of SAMS, with specifications for contamination control and certification of critical stages. The lack of a harmonized language for the definition of SAMS, coupled with a fragmented regulatory framework, presents a challenge for infection protection in pharmaceutical manufacturing. Key issues include the absence of specific regulations for stages prior to the introduction of SAMS and the lack of standards for inspections related to these stages. A desirable solution would be the mandatory extension of GMPs to the stages before SAMS introduction, with centralized control to ensure sterility and protection against infection throughout the entire manufacturing process.

## 1. Introduction

Starting Active Materials for Synthesis (SAMS) are defined as the raw materials or substances from which Active Pharmaceutical Ingredients (APIs) are derived or synthesized, marking the point at which Good Manufacturing Practices (GMP) begin to apply [1]. The pharmaceutical industry is increasingly recognizing the central role of SAMS in production processes. SAMS are an integral part of the formulation of pharmaceutical products that meet microbiological quality and strict safety requirements, thereby ensuring compliance with GMP [2,3]. This research document explores the importance of SAMS in pharmaceutical production, highlighting their impact on microbiological quality and safety, as well as the regulatory challenges associated with the implementation of control measures. SAMS serve as a fundamental component in the synthesis of APIs. Their quality directly affects the efficiency, safety, and conformity of the final product [4]. The purity of SAMS influences not only the pharmacological action of the drug but also impacts microbiological quality throughout the manufacturing process. This risk is exemplified by a documented case in which contamination of Starting Materials with *Acholeplasma laidlawii*, due to non-sterile tryptic soy broth, led to microbiological safety concerns and reinforced the FDA’s emphasis on upstream control [5]. Microbial contamination can have serious consequences, such as product recalls and public health crises, highlighting the importance of strict quality control measures in the selection and handling of SAMS [6]. Moreover, the microbiological quality of SAMS is essential for ensuring patient safety. The presence of contaminants, such as bacteria or fungi, can compromise the safety profile of pharmaceutical products. Recent studies emphasize the critical need for comprehensive methods for assessing biological contaminants in the biopharmaceutical manufacturing industry [7]. Effective biocontamination control strategies are essential to mitigate the risks associated with microbial contamination of SAMS, supporting the consistent production of safe and effective drugs [8]. However, the implementation of robust microbial quality control measures presents regulatory challenges. The pharmaceutical industry is governed by strict regulations that not only require verification of SAMS quality but also require continuous monitoring and validation processes to ensure ongoing GMP compliance [9]. Regulatory authorities such as the United States Food and Drug Administration (FDA) and the European Medicines Agency (EMA) have issued specific guidelines that stipulate the quality parameters necessary for SAMS, including identification, purity, and microbial load thresholds [10]. One of the main challenges associated with SAMS in pharmaceutical production is the transfer of materials from the research and development phases to commercial-scale manufacturing. Often, variations in microbial quality and characteristics of active materials arise during this transition. Ahsan et al. [11] underscore the importance of incorporating green synthetic chemistry approaches to improve the quality of SAMS while minimizing contamination risks during production. This is particularly critical in drug development involving natural or bioengineered components, where the inherent properties of starting materials may affect microbial stability [12]. The evolving landscape of SAMS presents opportunities for innovation, particularly through advances in nanotechnology and biocatalysis. Researchers have focused on the development of nanomaterials and their applications in pharmaceutical production [13,14]. The use of advanced materials can improve API stability and bioavailability while reducing the risk of biological contamination. For instance, biocatalysis facilitates the synthesis of complex pharmaceuticals in a more efficient and environmentally sustainable manner, leading to better microbiological quality [15,16]. As these novel manufacturing approaches emerge, it is essential to establish clear regulatory guidelines to evaluate the microbiological safety and compliance of new SAMS. In addition to regulatory issues, supply chain complexities further challenge the quality control of SAMS. Suppliers must provide verifiable documentation demonstrating adherence to hygiene and quality standards, especially when sourcing SAMS globally [17]. This is critical for preventing contamination issues from entering the pharmaceutical manufacturing process. Supplier audits and qualification are increasingly recognized as key components of a robust quality assurance framework, as emphasized by international regulatory guidance [18,19]. Integrating technological advances with microbial quality control practices can significantly mitigate the risks associated with SAMS in pharmaceutical production. Progress in microbial contamination detection enables rapid intervention and effective risk management [20]. These developments are essential for strengthening microbiological quality and facilitating GMP compliance. The use of natural plant-derived products in pharmaceutical synthesis further underlines the need for high-quality SAMS. While promising, such natural materials present unique microbiological contamination challenges [21]. As the industry increasingly embraces natural and semi-synthetic materials, rigorous quality assessment protocols must be in place to ensure safety and efficacy. In summary, the critical role of SAMS in pharmaceutical production cannot be overstated. Their influence extends beyond the initial synthesis of APIs to encompass microbiological quality and safety, both of which are vital to public health. The regulatory landscape presents ongoing challenges, particularly regarding the adoption of innovative materials and methods. Consequently, a comprehensive understanding of SAMS, combined with robust quality control and monitoring practices, is essential for navigating the complexities of pharmaceutical manufacturing while ensuring GMP compliance. The synthesis and application of SAMS hold promise for future pharmaceutical advancements, as the emphasis on quality and regulatory compliance becomes increasingly important. The integration of new technologies and sustainable practices within the SAMS framework can drive progress while safeguarding patient health and preserving pharmaceutical integrity [22,23]. As pharmaceutical manufacturing continues to evolve, the fundamental role of SAMS will remain central to ensuring the quality and safety of new therapeutic agents. In this work, we examine the regulatory approaches adopted by different international Regulatory Authorities concerning the selection and management of SAMS, with particular attention to microbiological control and infection prevention measures.

Therefore, the aim of this study is to analyze and compare current GMP guidelines issued by major international regulatory agencies, with a particular focus on identifying both differences and commonalities that may support the development of harmonized standards. Furthermore, the study explores the feasibility of extending GMP requirements to earlier stages of the manufacturing process, in order to ensure sterility and microbiological safety from the outset. In this context, two core research questions guide the analysis: (1) What are the main similarities and differences across GMP guidelines concerning microbiological control of SAMS, and how can these inform regulatory harmonization? (2) To what extent can GMP requirements be effectively extended upstream to enhance contamination control in pharmaceutical production?

## 2. Materials and Methods

This study involved an analysis of GMP guidelines issued by the following international Regulatory Agencies: the European Medicines Agency (EMA—Europe), the National Medical Products Administration (NMPA—China), the Federal Commission for the Protection against Sanitary Risk (COFEPRIS—Mexico), the Food and Drug Administration (FDA—United States), Health Canada (Canada), the Brazilian Health Regulatory Agency (ANVISA—Brazil), the Central Drugs Standard Control Organization (CDSCO—India), the Pharmaceutical Inspection Co-operation Scheme (PIC/S), which adopted the revised Annex 1 (August 2022) aligned with EMA [24], and the World Health Organization (WHO).

The regulatory documents and official guidelines were reviewed and compared with particular attention to the definition, identification, and handling of SAMS, as well as microbiological control measures applied in the stages preceding the final API manufacturing process.

Only regulatory documents published or revised between January 2015 and April 2025 were included. Inclusion criteria were: (1) official documents issued by international regulatory authorities; (2) availability in English or with an official English translation; and (3) explicit reference to microbiological control or GMP application to Starting Active Materials for Synthesis (SAMS). Exclusion criteria were: (1) unofficial or internal documents; (2) guidelines unrelated to SAMS or lacking microbiological provisions; and (3) documents not publicly accessible.

This comparative analysis aimed to identify regulatory differences, overlaps, and potential gaps in microbiological safety and contamination control measures across international agencies, with the goal of evaluating how effectively current GMPs mitigate microbial risks in the early phases of API production.

To ensure methodological transparency, a structured comparative approach was adopted. The analysis was guided by a matrix of predefined comparison criteria, including: (1) presence of specific guidelines for microbiological control of SAMS; (2) existence of measures prior to SAMS introduction; (3) application of risk-based Contamination Control Strategies (CCS); (4) requirements for supplier qualification; (5) presence of bioburden or sterility validation standards; and (6) regulatory clarity on when GMP obligations formally apply.

A qualitative content analysis of the regulatory documents was conducted by two independent reviewers. Each document was thematically coded using a deductive framework aligned with the comparison criteria. Discrepancies in interpretation were resolved through consensus discussion with a third researcher. The presence or absence of relevant provisions was recorded using binary (Yes/No) or categorical classifications (e.g., Partial, Encouraged).

Although the approach was primarily qualitative, structured data extraction and tabulation enabled the integration of quantitative summaries for visual comparison, as presented in Figure 1 and Table 1. This mixed-methods design facilitated a systematic comparison of regulatory frameworks and enhanced the robustness and reproducibility of the findings.

## 3. Results

An in-depth analysis of the international regulatory framework revealed a growing global consensus on the importance of ensuring microbiological safety and sterility of the raw materials used in the manufacture of APIs. Although there are regional differences in terminology, scope, and implementation, there is a clear trend: regulators are increasingly focusing on the need to extend contamination control strategies upstream, even before the formal application of GMPs. This section provides a detailed account of each regulatory agency’s position on the topics addressed in this study (Summary in Figure 1 and Table 1).

To better visualize the regulatory implications throughout the entire SAMS lifecycle, a dedicated flowchart (Figure 2) illustrates the critical control points (CCPs) where microbiological safety measures must be implemented. Each step is mapped in relation to major GMP guidelines, highlighting the convergence of international regulatory authorities on the importance of upstream contamination control.

### 3.1. European Union

The EMA imposes sterility requirements primarily through Annex 1 (2023) of the EU GMP, which prescribes a facility-wide contamination control strategy (CCS). The CCS must be integrated into the manufacturing environment, encompassing process design, facility layout, personnel behavior, and material flow. Barrier technologies, such as isolators and Restricted Access Barrier Systems (RABS), are required to minimize the risk of contamination during aseptic processing. These guidelines emphasize continuous environmental monitoring, regular personnel training, and process validation, particularly in cleanrooms classified as grades A to D. Notably, the sterility assurance process is considered holistic, focusing on upstream control beginning from the introduction of raw materials [25,26].

### 3.2. China

The NMPA has presented a revision of the GMP for sterile products (2025), closely aligned with the EU’s Annex 1. Risk-based CCS is required to deal with facility classification, material flow, and validation of aseptic techniques. The guidelines also encourage the use of quick alternative microbiological methods, provided they are appropriately validated. Furthermore, Chinese regulations emphasize early-stage contamination control, especially if the final API is intended for sterile use, in accordance with international standards for mitigating upstream microbiological risks [27].

### 3.3. Mexico

The COFEPRIS applies the principles of the GMP to sterilization within a general regulatory framework, without specific provisions for the introduction of materials. However, manufacturers are expected to maintain environmental control, personnel hygiene, and contamination risk management during sterile manufacturing. A significant challenge in the Mexican context is the limited infrastructure for routine microbiological surveillance and environmental monitoring, especially among smaller manufacturers. Inconsistent regulatory oversight, due to resource constraints, may result in inspection delays and variability in the enforcement of GMP standards [28,29].

### 3.4. United States

The U.S. FDA enforces sterility assurance in 21 Code of Federal Regulations (CFR) 210 and 211 sections and GMP guidelines, including sterile products produced by aseptic processing. The agency requires comprehensive environmental monitoring, aseptic technology validation, and media filling testing, which must be justified based on risk. The FDA explicitly states that International Organization for Standardization (IOS) cleaning room standards are insufficient in isolation and that microbiological controls must be integrated into a comprehensive GMP framework. Importantly, the FDA recognizes the risks of contamination prior to the formal application of GMP, as illustrated by a documented case involving *Acholeplasma laidlawii* contamination caused by non-sterile tryptic soy broth used in early manufacturing steps, as reported in the FDA’s CGMP Questions and Answers (Question 3, 2005) [5]. Regulations under 21 CFR 211.84(d)(6), along with international GMP guidelines [30,31,32,33], emphasize the importance of microbiological testing and val Giovanni Cangelosi idation at the component level, underlining the need for strict microbiological quality control beginning at the SAMS stage and even earlier.

### 3.5. Canada

Health Canada, through Guideline-0104 (GUI-0104) and related guidance, mandates that GMP control measures apply from the point of API and SAMS introduction and potentially even earlier when sterility is critical. The Canadian framework requires detailed microbiological characterization, impurity profiling, and supplier qualification, along with validated purification steps and contamination control strategies. Canada emphasizes the importance of bioburden assessment throughout the material lifecycle, including storage and transport, and promotes alignment with international standards such as International Council for Harmonisation (ICH) Q7 and Q11, adopting risk-based approaches to sterility assurance [34,35].

### 3.6. Brazil

Brazilian Resolução da Diretoria Colegiada (RDC) No. 301/2019, applied by ANVISA, is consistent with the structure of PIC/S and ICH Q7, but lacks clear definitions and microbiological control measures to start materials before the API is synthesized. No specific guidance is given on bioburden testing, aseptic processing validation, or pre-API microbiological risk assessment. While ANVISA aims to align with international norms, the agency faces operational challenges such as regional disparities in manufacturing oversight and limited frequency of audits. Past incidents involving substandard raw material imports have highlighted the need for more stringent upstream microbiological assessments [36].

### 3.7. India

India’s regulatory approach, led by the CDSCO, does not formally define SAMS or regulate early sterility controls. For the registration of APIs and formulations, specific documents such as GMP certificates, plant master files, and laboratory test results are required, but microbiological controls in the pre-API stages are inconsistently applied. The fragmented nature of the regulatory framework results partly from the diverse manufacturing scales across India, where small-to-medium enterprises may lack standardized microbiological control practices. Additionally, delays in policy updates and the lack of integration with digital tracking systems reduce transparency and hinder effective enforcement [37].

### 3.8. PIC/S

The Annex 1 of the Pharmaceutical Inspection Co-operation Scheme (PIC/S PE 009-17, 2023) provides a globally harmonized framework highlighting the need for microbiological control from the early stages of sterile production, including raw materials and production substances. The Pest Control Strategy (PCS) must include process design, environmental monitoring, equipment validation, and supplier qualification. The guidance states that it is insufficient to rely on final product tests and final sterilization and that sterilization must be guaranteed through process integration, certification, and preventive control. The classification of clean rooms (grades A–D), detailed microbial limits, and continuous monitoring are essential to maintain sterility during “residence” and “in operation”. PIC/S therefore sets a standard for the integration of GMP principles well before the formal API synthesis stage [38].

### 3.9. WHO

The WHO applies GMP requirements from the point of the API of SAMS, as per Technical Report Series (TRS) 957. Stages preceding SAMS fall outside the formal regulatory scope, though firms must define and justify the starting point of API production. The guidelines emphasise contamination prevention through validated facility design [e.g., Heating, Ventilation, and Air Conditioning (HVAC) systems], spatial segregation, and personnel control. Microbiological testing is required when relevant, including limits on total counts, endotoxins, and objectionable organisms, particularly for intermediates and biological products. Process validation and microbiological risk management extend to raw materials and environmental sources. While formal GMP begins at SAMS, WHO supports the early implementation of sterility controls and supplier qualification to prevent upstream microbial contamination [1,39].

## 4. Discussion

### 4.1. SAMS Definition and Management

SAMS play a central role in the manufacture of API, which defines the basics of the quality, safety, and efficacy of pharmaceutical products. SAMS are defined as raw materials or substances from which the API is derived or synthesized [38]. These may include natural products, chemical precursors, and other materials that undergo physical or chemical transformation throughout the manufacturing process. The importance of SAMS extends beyond their functional role in the production of drugs; they are essential to ensure compliance with regulatory frameworks that govern pharmaceutical manufacturing practices worldwide.

This analysis was performed to assess the impact of international regulations on the management of starting materials of active pharmaceutical ingredients, specifically in terms of sterility and microbiological safety [40]. Comparing regulatory frameworks across jurisdictions revealed a growing global consensus on the need to extend contamination control strategies upstream, well before the formal implementation of GMP. This shift reflects a greater awareness of the role of early microbiological risk mitigation to ensure the overall quality and safety of sterile drugs [41]. One of the main findings of this study is that several major regulatory authorities, including the European Union, China, the United States, and Canada, have developed or are actively implementing risk-based frameworks explicitly aimed at assessing the microbiological quality of raw materials and SAMS. In the European context, the revision of EMA Annex 1 (published August 2022, effective August 2023) introduces a CCS [41] to apply throughout the manufacturing process and extend to the first stages, including the handling of raw materials. Similarly, China’s draft GMP 2025 revision aligns with this direction and further advocates for validated alternative microbiological methods, highlighting the shift towards modernization and innovation in regulatory science. The identification of SAMS is decisive in the establishment of a reliable supply chain that responds to both qualitative and quantitative requirements. The identification of appropriate SAMs implies an in-depth characterization, including tests of identity, purity, potency, and stability [42]. Various protocols, such as high-performance liquid chromatography (HPLC) [40], mass spectrometry (MS), particularly LC-MS and LC-MS/MS [43,44], and nuclear magnetic resonance (NMR), are used to confirm the identity and assess the quality of these materials preventively. The quality of SAMs directly influences the final product, requiring a rigorous selection process that encompasses the validation of the geographic source, the qualification of suppliers, and an evaluation of materials based on historical quality performance. As international regulations emphasize the early mitigation of microbiological risks [44], the identification process has made a more in-depth examination, forcing manufacturers to adapt their analytical techniques to meet new standards that prioritize microbiological safety. The management of SAMS in a framework of GMP is essential to strengthen regulatory compliance and product integrity [45]. Management strategies must include all aspects of the SAMS life cycle, supply and storage to use, and elimination. Effective stock control systems can minimize the risk of contamination and degradation, guaranteeing the integrity of SAMS before their use in API synthesis. In addition, the integration of quality risk management principles into the SAMS management life cycle approach allows manufacturers to identify potential risks, implement preventive controls, and establish continuous monitoring strategies to mitigate microbiological risks [46].

The divergence between the EMA’s holistic CCS and the NMPA’s risk-based approach poses practical challenges for pharmaceutical manufacturers operating across multiple jurisdictions. Companies often need to implement dual or region-specific contamination control procedures to remain compliant with local regulations. This regulatory fragmentation increases operational complexity, delays product standardization, and generates additional costs related to validation, audits, and documentation [1,6]. Moreover, the absence of harmonized upstream microbiological controls complicates the development of a unified global quality strategy, requiring manufacturers to adapt workflows based on local regulatory expectations, ultimately hindering efficiency and scalability.

### 4.2. Microbiological Contamination Control

The recent emphasis on microbial control in pharmaceutical manufacturing has forced companies to critically assess their SAMS handling processes to improve the quality and safety of products. Microbiological safety in the context of the SAMS is particularly critical, given the critical implications of contaminated starting materials [47]. Regulatory organizations, such as the U.S. FDA’s Draft Guidance for Industry: Microbiological Quality Considerations in Non-Sterile Drug Manufacturing (September 2021) [48], and the EMA’s Annex 1 (2022) [41], have introduced strict guidelines to ensure that SAMS are free from pathogenic microorganisms. Bacterial contaminants, fungal spores, and other microorganisms can jeopardize the quality of APIs, resulting in serious consequences for patients. The advancement of microbiological test methods, particularly the introduction of rapid microbiological methods and biological detection assays [49], has become a central element of policy revisions influenced by international directives. These innovations offer manufacturers the ability to perform more effective microbiological risk assessments, adapt their validation protocols accordingly, and implement real-time test methods throughout the production process, aligning with the transition to a preventive approach to GMP regulations.

However, implementing these alternative microbiological methods in resource-limited settings presents significant challenges. High initial investment costs, lack of technical expertise, inadequate infrastructure, and insufficient regulatory guidance can hinder adoption. In many low- and middle-income countries, traditional microbiological methods remain predominant due to their lower cost and regulatory familiarity, despite limitations in sensitivity and turnaround time. Addressing these barriers is essential to ensure equitable global access to improved microbiological control systems [50,51].

Various regulatory authorities across jurisdictions also guide the management and testing of SAMS [52]. Although the emphasis on security and microbiological quality is universally recognized, regulations can vary considerably in their specific requirements and the degree of rigor. For example, the Japanese Pharmaceutical and Medical Devices Agency (PMDA) has distinct guidelines that can diverge from those established by the FDA and EMA, requiring acute awareness of local regulatory practices. In addition, the evolution of the dynamics of the global market encourages manufacturers to adapt to these regulatory differences while retaining compliance with global quality and safety principles. The role of harmonization initiatives, such as the guidelines of the ICH, in filling these regulatory gaps cannot be underestimated because they promote a standardized environment conducive to commercial insurance and global security. The SAMS serve as the cornerstone of API manufacturing, where their definition, identification, management, and microbiological safety are essential to meet both product and regulatory requirements. With international regulations increasingly emphasizing early microbiological risks [52], manufacturers are forced to innovate and adopt advanced analytical techniques aimed at ensuring the safety and quality of SAMS. Thanks to the diligent management of SAMS in accordance with various regulatory frameworks, pharmaceutical companies can both comply with their compliance obligations and enhance therapeutic efficiency and the safety of their products. The interaction between SAMS and regulatory membership underpins a constantly evolving landscape, requiring a proactive approach that anticipates regulatory revisions while promoting public health thanks to meticulously produced pharmaceutical products.

### 4.3. Future Perspectives

In the future, regulatory developments are likely to continue focusing on enhancing the control and management of microbiological risks throughout the pharmaceutical manufacturing lifecycle, especially given the growing complexity of products such as biologics and cell therapies [53]. Progress in rapid and alternative microbiological testing methods, supported by internationally validated protocols, is expected to play an increasingly critical role in improving real-time contamination detection and process control [54]. For instance, PCR-based assays, ATP bioluminescence detection, and biosensor technologies are being actively evaluated for their potential to provide faster and more sensitive microbiological monitoring compared to traditional culture-based methods [20]. These tools could significantly enhance sterility assurance, particularly during critical stages of manufacturing. Moreover, greater global regulatory convergence will be essential to standardise expectations and ensure equitable access to high-quality sterile drugs across different markets. Concrete harmonization efforts are being led by the International Council for Harmonization (ICH) through guidelines such as ICH Q7 (Good Manufacturing Practice for Active Pharmaceutical Ingredients), ICH Q9 (Quality Risk Management), ICH Q10 (Pharmaceutical Quality System), and ICH Q11 (Development and Manufacture of Drug Substances). These guidelines provide a structured foundation for aligning microbiological control strategies at the international level [55,56,57,58]. Collaborative initiatives involving WHO, ICH, PIC/S, and regional authorities are anticipated to drive harmonisation efforts [59], particularly in promoting the extension of GMP standards to stages preceding the formal introduction of starting materials.

### 4.4. Implications for Clinical Practice

Although this analysis focuses primarily on manufacturing and regulatory aspects, its implications for clinical practice are significant. The microbiological quality of SAMS directly affects the safety and therapeutic efficacy of pharmaceuticals administered to patients, especially in vulnerable populations such as pregnant women, oncology patients, and immunocompromised individuals [60]. Contamination incidents, even when originating from early stages such as SAMS production, can cause therapeutic failures, product recalls, and serious adverse events [61,62]. Thus, robust upstream control systems not only safeguard the microbiological integrity of pharmaceutical products but also reinforce the trust of clinicians and patients in the safety and effectiveness of treatments in all parts of clinical pathways, particularly in surgery and chronic care setting [63,64,65,66].

### 4.5. Limitations of the Study

This study is based on publicly available regulatory documents, guidelines, and interpretations current at the time of the analysis. It does not include unpublished, non-official, or country-specific regulatory practices that might exist outside the major regulatory frameworks reviewed. Despite efforts to include a variety of jurisdictions, limited availability of documents in English and sparse resources from lower- and middle-income countries may have led to an underrepresentation of certain regulatory landscapes. Additionally, some major regulatory agencies—such as the Pharmaceuticals and Medical Devices Agency (PMDA, Japan) and the Therapeutic Goods Administration (TGA, Australia)—were not included in the comparative analysis. While documents from the TGA are available in English, neither agency publishes independent microbiological guidelines specific to SAMS, distinct from the international standards already referenced in this study (e.g., ICH, PIC/S, WHO). Therefore, their regulatory positions were considered sufficiently represented through these overarching frameworks. Furthermore, given the dynamic and evolving nature of regulatory systems, some frameworks might evolve in the near future, possibly altering their current alignment with international best practices.

## 5. Conclusions

The lack of a unified definition of SAMS and the fragmented regulatory landscape pose significant challenges to microbiological safety in pharmaceutical manufacturing. Key issues include the absence of specific guidelines for stages prior to the introduction of SAMS, as well as the lack of standardised inspection protocols for these earlier phases. To address these challenges, it is crucial to extend GMP to cover all upstream stages before the introduction of SAMS. A global harmonisation of regulatory frameworks, with a focus on microbiological quality control, would enhance sterility assurance throughout the entire manufacturing process. This would ultimately safeguard public health by ensuring the safety, quality, and efficacy of pharmaceutical products, reducing the risk of contamination, and improving infection control strategies worldwide.

## Figures and Tables

**Figure 1 microorganisms-13-01595-f001:**
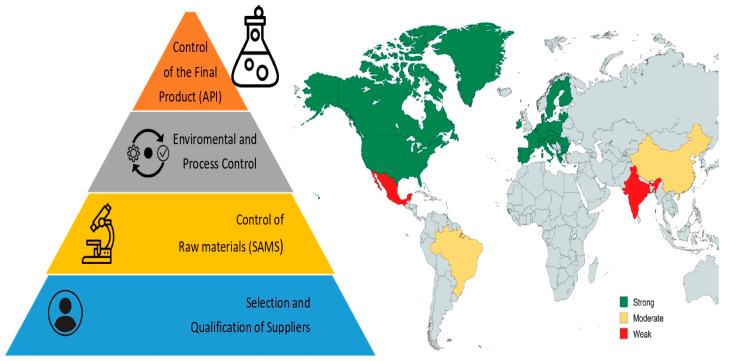
SAMS: Microbiological Oversight and Regulatory Frameworks.

**Figure 2 microorganisms-13-01595-f002:**
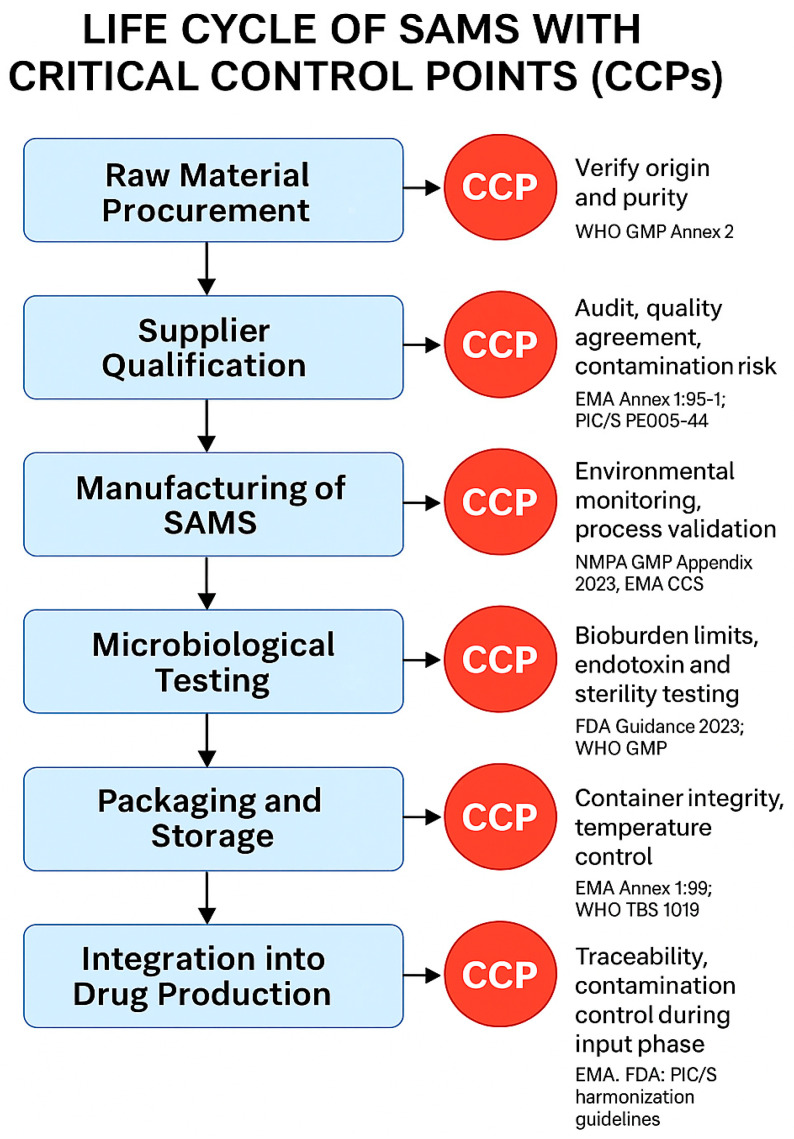
Flowchart illustrating the lifecycle of SAMS, highlighting CCPs and their alignment with key international regulatory requirements. Each stage represents a critical area for microbiological risk control.

**Table 1 microorganisms-13-01595-t001:** Microbiological Control and Regulatory Approaches to SAMS.

Condition/Institution	EMA (Europe)	NMPA (China)	COFEPRIS (Mexico)	FDA (USA)	Health Canada (Canada)	ANVISA (Brazil)	CDSCO (India)	PIC/S	WHO
Guideline for microbiological control of SAMS	Yes	Yes	No	Yes	Yes	Partial	No	Yes	Yes
Measures before SAMS introduction	Yes	Yes	No	Yes	Yes	No	No	Yes	Partial
Requirement for risk-based CCS	Yes	Yes	No	Yes	Yes	No	No	Yes	Partial
Control of upstream stages (before GMP application)	Yes	Yes *	No	Yes	Yes	No	No	Yes	Encouraged
Supplier qualification (microbiological quality)	Yes	Yes	No	Yes	Yes	Partial	No	Yes	Yes
Bioburden/sterility validation standards	Yes	Yes	No	Yes	Yes	No	No	Yes	Partial
Clear definition of GMP starting point	Yes ^1^	Yes	No	Yes	Yes	No	No	Yes	Yes ^1^
Emphasis on rapid/alternative microbiological methods	Yes	Yes	No	Yes	Partial	No	No	Yes	Partial
Mandatory environmental and personnel microbiological monitoring	Yes	Yes	Partial	Yes	Yes	No	No	Yes	Partial
Upstream synthetic process description required	N/A	Optional	No	Yes	Yes	Yes	Yes	N/A	N/A

**Legend.** EMA: European Medicines Agency; NMPA: National Medical Products Administration; COFEPRIS: Federal Commission for the Protection against Sanitary Risk; FDA: Food and Drug Administration; ANVISA: Brazilian Health Regulatory Agency; CDSCO: Central Drugs Standard Control Organization; PIC/S: Pharmaceutical Inspection Co-operation Scheme; WHO: World Health Organization; SAMS: Starting Active Materials for Synthesis; GMP: Good Manufacturing Practices; CCS: Contamination Control Strategy; Health Canada: Canadian Health Regulatory Authority; **Table values:** Yes: Clearly required; **Yes** *: Present as a recommended (not mandatory) requirement; Yes ^1^ = Required starting from the point of SAMS introduction; No: Not required or not mentioned; Partial: Mentioned but not comprehensive or consistent Optional = Mentioned as useful, but not required; Encouraged: Not mandatory, but explicitly recommended; N/A = Not applicable in the regulatory framework.

## Data Availability

No new data were created or analyzed in this study. Data sharing is not applicable to this article.

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
