# Peer review of "Preventing Microorganism Contamination in Starting Active Materials for Synthesis from Global Regulatory Agencies: Overview for Public Health Implications"

_microorganisms, 2025, doi:10.3390/microorganisms13071595_

Round 1
Reviewer 1 Report
Comments and Suggestions for Authors
The manuscript evaluates regulatory differences across countries regarding Starting Active Materials for Synthesis (SAMS). The subject is relevant and the approach is sound. However, I have a few comments that the authors could address to improve the manuscript’s clarity and quality.
Introduction: Include a brief sentence defining or explaining the concept of SAMS to readers unfamiliar with the term.
L78: The authors mention risks of products recall due to microbiological contamination. Could the authors provide an example (for instance, a documented incident involving SAMS contamination).
General comment: Sometimes the authors use SAMS (capital) and in others, they use SAMs (lower case). The same comment holds for British or American English (e.g., standardise or standardize). Please verify throughout the text.
The authors cite a contamination case involving Acholeplasma laidlawii. Clarify whether this case is included in references [27-30] or provide a specific citation.
The analysis of agencies like EMA, FDA, and PIC/S is detailed, whereas COFEPRIS, ANVISA, and CDSCO are described generically. To balance the comparison, the authors should include further details of these agencies, such as specific barriers (lack of infrastructure, for instance) or examples of regulatory failures.
The discussion blends topics on SAMS characterization and microbiological contamination, which may confuse readers. I suggest splitting the discussion into clearer subsections, such as “SAMS Definition and Management” and “Microbiological Contamination Control” to improve clarity.
The methods section lacks details regarding the comparison criteria, the analytical tools adopted, and the qualitative/quantitative approach. To improve transparency, this section should be supplemented.
The future perspectives section is broad, but lacks specific examples of emerging technologies or harmonization initiatives. The authors should include concrete examples, such as PCR-based testing or specific ICH guidelines.
Author Response
Dear Reviewer,
We hope you will fully appreciate this new version, shaped by your valuable suggestions.
The Authors

Reviewer 2 Report
Comments and Suggestions for Authors
The manuscript provides a comprehensive comparative analysis of international regulatory frameworks (EMA, NMPA, FDA, etc.) regarding microbiological safety in SAMS and API manufacturing. Obatained results address a critical gap in understanding regulatory approaches to microbiological safety in API production. With revisions to strengthen methodology, terminology, and public health linkages, it has the potential to inform both policy and industry practices. The specific comments were shown below:
1. Detail the criteria used for selecting and comparing regulatory documents (e.g., inclusion/exclusion criteria, temporal scope).
2. Address how discrepancies in "upstream contamination control" (e.g., EMA’s holistic CCS vs. NMPA’s risk-based approach) impact manufacturers operating in multiple jurisdictions.
3. Discuss the practical challenges of implementing "alternative microbiological methods" (mentioned in the NMPA section) in resource-limited settings.
4. Reformulate Table 1 to improve readability.
5. Include a flowchart mapping the SAMS lifecycle with critical control points aligned to regulatory requirements.
5. Clarify the rationale for excluding certain agencies (e.g., PMDA Japan, TGA Australia) from the analysis.
6. Update references to reflect recent guidelines.
Author Response
Dear Peer Reviewer,
We hope you will fully appreciate this new version, shaped by your valuable suggestions.
The Authors

Reviewer 3 Report
Comments and Suggestions for Authors
The present study investigates the regulations established by a number of international regulatory agencies, with a particular emphasis on microbiological control and infection control measures employed in the production process of Starting Active Materials (SАМS).
The manuscript is well written and structured. It corresponds to the scope of the journal Microorganisms. But some corrections are required.
Point 1: The abstract should be an objective representation of the results. For example, on page 2, lines 44-46, it is stated that the regulations in China, Mexico, and India have notable gaps, failing to provide clear guidelines on SAMS sterility and protection against infectious contamination. However, the results described in subsection 3.2 and the data in Table 1 show that China has developed and is actively implementing risk-based approaches that directly address the microbiological quality assessment of SAMS.
Point 2: It is not recommended to separate Section 1.1, but rather to combine it with the introduction. The purpose and objectives should be written in the traditional manner.
Point 3: Figure 1 and Table 1 should be inserted into the main text close to their first citation.
Author Response

(The authors gave the same response as above.)

Round 2
Reviewer 2 Report
Comments and Suggestions for Authors
The revised version was not shown in this round, and major revision was suggested following resubmission.
Author Response
Dear Peer Reviewer,
Sorry for the oversight. We re-uploaded the revised version.
We hope you will fully appreciate this new version, shaped by your valuable suggestions.
The Authors

Round 3
Reviewer 2 Report
Comments and Suggestions for Authors
The MS was revised according to the proposed comments and an acceptance decision was suggested.